# Multimodal Analgesia (MMA) Versus Patient-Controlled Analgesia (PCA) for One or Two-Level Posterior Lumbar Fusion Surgery

**DOI:** 10.3390/jcm9041087

**Published:** 2020-04-11

**Authors:** Sung-Woo Choi, Hyeung-Kyu Cho, Suyeon Park, Jae Hwa Yoo, Jae Chul Lee, Min Jung Baek, Hae-Dong Jang, Joong Suk Cha, Byung-Joon Shin

**Affiliations:** 1Department of Orthopaedic Surgery, Soonchunhyang University College of Medicine, Seoul 04401, Korea; shinestellar@naver.com (H.-K.C.); jlee@schmc.ac.kr (J.C.L.); figojun@naver.com (J.S.C.); 2Department of Biostatistics, Soonchunhyang University College of Medicine, Seoul 04401, Korea; suyeon1002@schmc.ac.kr; 3Department of Anesthesiology and Pain Medicine, Soonchunhyang University College of Medicine, Seoul 04401, Korea; 89523@schmc.ac.kr; 4Department of Obstetrics and Gynecology, Bundang CHA Hospital, Seongnam 13496, Korea; goodgood75@naver.com; 5Department of Orthopaedic Surgery, Soonchunhyang University College of Medicine, Bucheon 14584, Korea; khaki00@schmc.ac.kr

**Keywords:** multimodal treatment, analgesia, patient-controlled, lumbar vertebra, arthrodesis, pain, postoperative, length of stay, cost, preemptive

## Abstract

A multimodal analgesic method was known to avoid the high-dose requirements and dose-dependent adverse events of opioids, and to achieve synergistic effects. The purpose of this study was to compare the efficacy of our multimodal analgesia (MMA) regimen with that of the patient-controlled analgesia (PCA) method for acute postoperative pain management. Patients who underwent one or two-level posterior lumbar fusion (PLF) followed by either MMA or PCA administration at our hospital were compared for pain score, additional opioid and non-opioid consumption, side effects, length of hospital stay, cost of pain control, and patient satisfaction. From 2016 through 2017, a total 146 of patients were screened. After propensity score matching, 66 remained in the PCA and 34 in the MMA group. Compared with the PCA group, the MMA group had a shorter length of hospital stay (median (interquartile range): 7 days (5–8) vs. 8 (7–11); *P* = 0.001) and lower cost of pain control (70.6 ± 0.9 USD vs. 173.4 ± 3.3, *P* < 0.001). Baseline data, clinical characteristics, pain score, additional non-opioid consumption, side effects, and patient subjective satisfaction score were similar between the two groups. The MMA seems to be a good alternative to the PCA after one or two-level PLF.

## 1. Introduction

Effective postoperative pain control allows for faster recovery, reduced complications, and improved patient satisfaction. This is common knowledge among clinicians. However, approximately more than 80% of patients who undergo surgical procedures experience acute postoperative pain, and 75% of those with postoperative pain report moderate, high, or extreme pain severity [1,2,3]. Inadequately controlled pain increases the stress response in a way that affects the immune system, which leads to delayed healing and is a known risk factor of persistent postsurgical pain [4,5,6,7].

Spinal surgeries are generally associated with intense pain in the postoperative period, especially for the initial few days [8]. Traditionally, pain management after spine surgery relies heavily on opioids [9,10]. An opioid has strong effects on pain control but also has many side effects. However, as the frequency of the use of prescription opioid analgesics more than doubled between 2001 and 2013 in the world and in many countries, the incidence of opioid misuse is escalating, causing thousands of deaths annually [11,12].

One common technique to address pain after surgery is patient-controlled analgesia (PCA) [13,14]. PCA refers to any method that allows the patient to self-administer intravenous narcotic medication. Although highly effective at relieving pain, narcotic analgesics can have severe side effects, including nausea, vomiting, confusion, constipation, urinary retention, dizziness, sedation, respiratory depression, and pruritus [15,16]. Multimodal analgesia (MMA) is a good alternative to PCA. The concept of MMA was introduced more than two decades ago as a technique to improve analgesia and reduce the incidence of opioid-related adverse events [17]. MMA uses a combination of analgesic drugs with different mechanisms of action designed to promote effective pain control, decrease opioid consumption, and reduce medication-related side effects [15,18]. As reported in many studies, the MMA protocol varies between authors [4,15,18,19,20]. The purpose of this study was to compare the efficacy of our MMA protocol, which is focused on preemptive analgesia (oral medication and bupivacaine injection before the surgery), with the conventional intravenous PCA method.

## 2. Methods

### 2.1. Patients

A retrospective review of a prospective collected spine registry at our institution was conducted to compare the effectiveness of our MMA protocol and the conventional PCA protocol for postoperative pain management. The study was approved by the Soonchunhyang University of Institutional Review Board (approval No. 2017-03-002-001). Patients were included if they underwent a one-level or two-level posterior lumbar fusion (PLF) surgery with PCA and MMA between June 2016 and July 2017. The exclusion criteria were: any history of prior lumbar spine surgery, hepatic dysfunction, renal insufficiency, depressive disorder, an emergency surgery, a history of drug/alcohol abuse, or prior adverse or allergic reactions to any of the analgesic medications to be administered (celecoxib, pregabalin, acetaminophen, or oxycodone). In our institution, the postoperative pain control method switched from a PCA-based to MMA-based regimen from January 2017. One hundred and nine PCA patients were collected in 2016 and 37 consecutive patients enrolled in 2017.

### 2.2. Study Design

All patients who had a history of more than three months of conservative pain control underwent one or two-level posterior lumbar fusion (PLF) with degenerative spinal disorder. The two protocols are described in detail in Table 1. The preemptive multimodal analgesia was composed of celecoxib, acetaminophen, pregabalin, and oxycodone. Only in the MMA group, all the agents were administered around 1 h before surgery, and a local anesthesia was injected prior to skin incision. Both groups had general anesthesia with propofol and remifentanil. After the surgery, patients in the PCA group were connected to a patient-controlled analgesia pump and received a continuous intravenous infusion of 1 mL/h with additional patient-controlled bolus doses of 1 mL with a lock-out time of 5 min. Patients in the MMA group were injected with tramadol 50 mg intravenously. When they woke up in the anesthesia, MMA medication was administrated. When an additional rescue regimen was needed, non-opioids were used first. If these did not result in significant pain reduction, an opioid was used. The MMA protocol in this study was developed by a team of anesthesiologists and surgeons at our institution. Similar protocols have been used in other studies within the field of spine surgery [15,19] and in other fields [21].

### 2.3. Outcome Measures

For each patient, baseline data and clinical characteristics were investigated. Baseline data included age, sex, current smoking status, and body mass index (BMI). Clinical characteristics included preoperative pain on the 10-point numeric rating scale (NRS), Oswestry Disability Index (ODI), preoperative pain medication, American Society of Anesthesiologist (ASA) physical class, autologous iliac bone graft, suction drain use, intraoperative complications, and operation time. The pain position was classified into an axial, radiating to a leg, and a combined. When NRS was more than 4 points, it was regarded as significant. The preoperative pain medication was defined from one month before surgery and assorted into five groups: none, acetaminophen/nonsteroidal anti-inflammatory drug (NSAID), weak-opioids, strong-opioids, combination method. The combination is a complex method such as a steroidal injection plus oral medication and/or other type of pain medication used together. The ASA physical class was estimated by an anesthesiologist before the preoperative evaluation. Postoperative NRS scores were recorded during assessments at 1, 4, 8 h after surgery on postoperative day (POD) 0 and every 8 h on POD 1, 2. The patients were asked by one of the authors to rate their average pain on the NRS (0–10). An average NRS was then recorded for each postoperative day. Additional opioid consumption during the inpatient stay was calculated for each patient and expressed in oral morphine equivalents (ME) per day. For statistical comparisons, the oral morphine equivalent dose was calculated based on the following oral ratio: pethidine (1:3). Additional non-opioid analgesia consumption using IV tramadol during the inpatient stay was calculated for each patient and expressed in mg/day. Patients were interviewed about nausea/vomiting and abdominal discomfort at 1, 4, 8 h after surgery on postoperative day (POD) 0 and every 8 h on POD 1, 2. Additional antiemetic drug, IV ramosetron (0.3 mg) was administered if needed. Length of hospital stay and the cost of the pain control (PCA, MMA, and additional rescue regimen) were documented. The cost of the PCA group consisted of setup costs, which consisted of machine rental costs and installation costs, maintenance costs per used day, medicine costs, which mixed in the PCA pump and costs of the rescue regimen. Additionally, the cost of the MMA group consisted of costs of the regimen prescribed by the MMA protocol and costs of the rescue regimen. The patient’s satisfaction with their pain control method was assessed on POD 3 using a 5-point categorical scale (1: very satisfied, 2: somewhat satisfied, 3: neutral, 4: somewhat dissatisfied, 5: very dissatisfied).

### 2.4. Statistical Analysis

Statistical analyses were conducted using SPSS version 26.0 (SPSS, Chicago, IL). Continuous variables were compared using the student t-test or Mann–Whitney test. Nominal variables were compared using Pearson’s chi-squared test or Fisher’s exact test. A *P*-value less than 0.05 denoted statistical significance and the tests performed were two-tailed. Because this study was not a randomized trial, the propensity score matching (PSM) method was used to reduce selection bias between two groups (‘Matchit’ packages, R version 3.1.2.). In Table 2, other variables except age and BMI were not different between the MMA and PCA group. The propensity score was calculated using age and BMI and 1:2 matching was performed [22,23,24,25].

## 3. Results

Thirty-seven patients in the MMA group received our multimodal pain management regimen and 109 patients in the PCA group received conventional pain management. Twelve patients were withdrawn from the study. Three patients in the MMA group wanted to change to PCA due to severe pain. Nine patients in the PCA group discontinued their PCA due to uncontrolled severe nausea/vomiting. Then, the patients were matched on the basis of age and BMI in a 1:2 ratio to reduce selection bias between two groups by using the propensity score matching (PSM) method. One hundred patients were analyzed in this study. Of these, 34 (34%) received MMA and 66 (66%) received PCA (Figure 1). The baseline data and clinical characteristics were similar between groups (Table 2). No major identifiable intraoperative and postoperative complications such as neural injury, postoperative wound infection, were observed in either treatment group. In the PCA group, 12 patients terminated IV PCA pump on POD 1 and 38 patients terminated on POD 2 and 16 patients terminated on POD 3.

### 3.1. Pain Score And Additional Pain Control Medication

The mean NRS was not significantly different between the groups (Table 3). Additional opioid use trended toward lower use in the MMA group than the PCA group at POD 0 (2.7 ± 6.7 vs. 5.1 ± 8.3, *P* = 0.137) and at POD 1 (2.2 ± 5.9 vs. 8.2 ± 16.1, *P* = 0.059). However, these trends did not reach statistical significance. At POD 2, the MMA group had lower opioid use than the PCA group (1.5 ± 4.8 vs. 7.7 ± 14.5, *P* = 0.014) (Table 3). Additional non-opioid use was similar between the two groups (Table 3).

### 3.2. Side-Effects

The use of medication for postoperative nausea and vomiting (PONV) was similar between the MMA and PCA groups (Table 4). There was no difference in the rate of postoperative abdominal discomfort (26.5% vs. 22.7% on POD 0, 23.5% vs. 18.2% on POD 1, 2.9% vs. 9.1% on POD 2) (Table 4).

### 3.3. Length of Hospital Stay, Cost, and Patient Satisfaction

Length of hospital stay was significantly shorter in the MMA group compared to the PCA group (7 [5,6,7,8] vs. 8 [7,8,9,10,11], median [interquartile range], *P* = 0.001) (Table 5). The cost of the MMA treatment was less than the PCA treatment (USD 70.6 ± 0.9 vs. 173.4 ± 3.3, *P* < 0.001) (Table 5). The subjective satisfaction score was similar between the two groups (Table 5).

## 4. Discussion

The NRS, additional opioid and non-opioid rescue medication were not different between the groups on POD 0 and 1. Moreover, there was less consumption of additional opioids in the MMA group compared to the PCA group on POD 2. These results mean that the pain control effect of our MMA protocol is not inferior to that of the convention PCA protocol. In the 1970s, the development of PCA was expected to achieve rapid and complete pain control after surgery. However, there is a possibility that patients will self-administer the pain medication to get high, and therefore, use large amounts of opioids. In addition, opioids have been reported to have a high frequency of side effects such as nausea and vomiting [26].

A multimodal approach to postoperative pain is attractive for a number of reasons [27]. First, the use of non-opioid analgesia, either alone or in combination with opioids, may reduce the patient’s level of pain and overall postoperative opioid requirements. Second, the use of non-opioids may spare patients from potentially devastating adverse effects, such as respiratory distress and death, which are associated with excessive opioid administration [28]. Third, the use of non-opioid treatment strategies may decrease the incidence of opioid-related side effects such as somnolence, confusion, urinary retention, ileus, delayed time to oral intake, and delayed patient mobilization [20].

The use of MMA regimens after joint arthroplasty has gained acceptance [29,30], but its use in the field of spine surgery is limited. A growing body of evidence supports the use of multimodal analgesia for spine surgery [4,9,31,32]. Several investigations have been conducted regarding the use of MMA as compared with the conventional pain control methods. In an randomized controlled study by Garcia et al. [20], patients who underwent lumbar decompression received intravenous morphine only or a preemptive MMA regimen (preoperative celecoxib, pregabalin, and extended-release oxycodone, and Marcaine injection prior to wound closure). The MMA group had a lower visual analog scale (VAS) pain score and less morphine consumption. Singh et al. [15] retrospectively compared an MMA regimen (preoperative cyclobenzaprine, pregabalin, and oxycodone, and intraoperative IV acetaminophen, dexamethasone, fentanyl, and ketamine) with a PCA regimen in patients who underwent a single-level minimally invasive transforaminal lumbar interbody fusion (MIS TLIF). The MMA group had reduced hospital narcotic consumption, nausea/vomiting, and length of hospital stay. However, the analgesic effect for pain control is similar between MMA and PCA. In a retrospective study of multilevel instrumentation [19], authors used MMA medication (celecoxib, acetaminophen, and gabapentin) and inserted an epidural catheter and injected bupivacaine postoperatively. The MMA group showed improved pain control, reduced opioid consumption, and earlier mobilization. Although epidural catheter insertion is an effective analgesic method, it may worsen cases with complications. Kim et al. [18] compared a preemptive MMA regimen (celecoxib, pregabalin, acetaminophen, and extended-release oxycodone) with an intravenous morphine regimen in patients who underwent a single-level lumbar fusion. The MMA group had lower VAS and ODI scores at all times, except the ODI score on postoperative day 1. However, no significant differences were found in intraoperative blood loss, postoperative suction drain output, and nonunion rate.

In the present study, our MMA regimen was focused on preemptive analgesia, which is administered before pain onset and based on central sensitization. It composed of MMA medication (celecoxib, acetaminophen, pregabalin, and opioid) and long-acting local analgesic (bupivacaine) injection before pain stimuli. Celecoxib, which is a selective inhibitor of cyclooxygenase (COX)-2, inhibits prostaglandin synthesis in the spinal cord and in the periphery so it diminishes the hyperalgesic state after surgical trauma [18]. It is associated with reduced opioid requirements after surgery, and some studies reported lower postoperative pain scores [1,33,34]. Acetaminophen is a popular choice for managing mild-to-moderate pain, and it can induce nonsteroidal anti-inflammatory (NSAID)- and opioid-sparing roles [18,35]. In addition, pregabalin prevents the release of nociceptive neurotransmitters in the spinal cord, so it may limit opioid-induced hyperalgesia [36]. Opioids act their analgesic activity through opioid receptors located in the central nervous system. These medications (NSAIDs, acetaminophen, pregabalin/ gabapentin, opioids) have shown a successful combination [1]. Additionally, the role of local anesthetic is reducing postoperative wound hyperalgesia [36].

These data show that additional opioid use was less in the MMA group compared to the PCA group on POD 2. This may be related to PCA (38/66, 57%) that terminated on POD 2 in the PCA group. A relatively regular infused painkiller in the PCA was abruptly stopped, and it seemed that the rescue medication may be necessary. Some literature reported pain is one of the most common causes of delayed discharge [37]. In the current study, although the NRS was similar, the MMA group discharged more quickly than the PCA group. The PCA is relatively more expansive than MMA regimen because it consists of machine equipment rental cost, clinician management cost, and including drugs cost. In the present study, the MMA therapy is approximately seven times cheaper than PCA. MMA therapy is approximately seven times cheaper than PCA. However, both groups had similar overall satisfaction with pain control at the end of the acute postoperative phase.

This study had several limitations. First, our study was not randomized. So, there may have been selection bias with respect to which patients were assigned to each pain control method. However, we used the PSM method to reduce selection bias between the two groups. Ultimately, the patients in the MMA group did not differ in most demographic variables from those in the PCA group. Second, our study was a small sample size. Although rescue opioid usage tended to be lower in the MMA group, the difference did not reach statistical significance. In addition, although the regimen for each MMA protocol is different, the side effects with MMA are typically less common than with PCA [15,16,19]. We had hypothesized that postoperative nausea and vomiting (PONV) would be less common in the MMA group. However, PONV in the two groups was similar, even though opioid use was less in the MMA group on POD 2. We think that this mismatch may be caused by the progressing PCA regimen in our institution and the sample size. The PCA regimen is improving, also owing to its known side effects such as nausea/vomiting. The PCA regimen of this study includes two types of antiemetics (ramosetron and palonosetron) preemptively. Although others may suppose that this (PCA contained 2 antiemetics vs. MMA contained 1) is unfair, we think that this reflects the real clinical situation. Third, we could not investigate the rehabilitation schedule of the patients. The present study demonstrated that our MMA protocol was not inferior to the conventional PCA method for postoperative pain control after a one or two-level posterior lumbar fusion surgery. Further studies are needed to validate the results of this study.

In this study, multimodal analgesia, which uses a combination of selective COX-2 inhibitors, pregabalin, long-acting opioids, and acetaminophen appears to be a safe and effective analgesic regimen after one or two-level posterior lumbar fusion surgery. Additional opioid use was lower in the MMA group compared to the PCA group on POD 2. Moreover, patients who received MMA did not report any increase in their pain compared to those patients receiving PCA. Moreover, the MMA group discharged more quickly and at a lower total cost than the PCA group.

## 5. Conclusions

The present study demonstrated that our MMA protocol was not inferior to the conventional PCA method for postoperative pain control after a one-level or two-level posterior lumbar fusion surgery. MMA was beneficial in terms of cost and length of hospital stay. MMA seems to be a good alternative to PCA after short level posterior lumbar fusion.

## Figures and Tables

**Figure 1 jcm-09-01087-f001:**
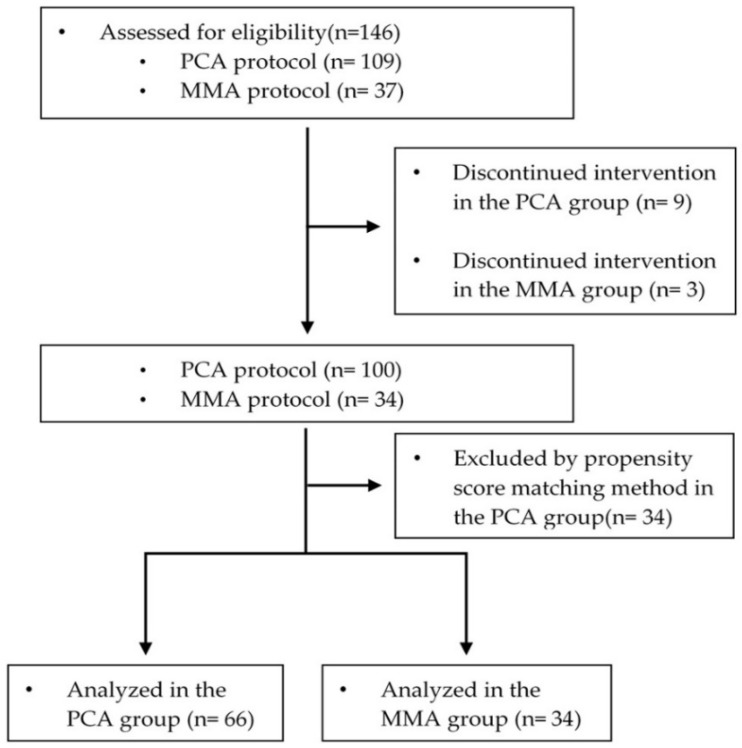
Study design and flowchart.

**Table 1 jcm-09-01087-t001:** Patient controlled analgesia (PCA) vs. Multimodal analgesia (MMA) protocol.

		PCA	MMA
Preoperative		No intervention	Celecoxib 200 mg PO
			Acetaminophen 650 mg PO
			Pregabalin 75 mg PO
			Oxycodone 10 mg PO
			Dexamethasone 10 mg IV
			Ramosetron 0.3 mg IV
Intraoperative	Anesthesia *	Induction—Propofol	
		Maintenance—Desflurane, Remifentanil	
		Recovery—Fentanyl	
	Surgery	No intervention	Bupivacaine + Epinephrine 10 mL SQ ** (prior to incision)
Postoperative Day 0	Recovery room	1. PCA keep	1. Tramadol 50 mg IV
		Fentanyl 1.5 mg IV	
		Nefopam 100 mg IV	
		Propacetamol 5 g IV	
		Ramosetron 0.6 mg IV	
		Palonosetron 0.075 mg IV	
		2. Fentanyl IV (PRN)	2. Fentanyl (PRN)
	In-patient room	1. PCA keep	1. MMA PO medication
			Celecoxib 100 mg PO, 1-tab q12h
			Acetaminophen 650 mg PO, 1-tab q8h
			Pregabalin 75 mg PO, 1-tab q12h
			Oxycodone/Naloxone 10 mg PO, 1-tab q6h
		2. PRN ***	2. PRN ***
			3. Ramosetron 0.9 mg IV
Postoperative Day 1,2		1. PCA keep	1. MMA PO medication
		2. Routine fluid ****	2. Routine fluid ****
		3. PRN ***	3. PRN ***

PCA = patient-controlled analgesia; MMA = multimodal analgesia; PO = per os; IV = intravenous; SQ = subcutaneous; PRN = pro re nata * Anesthesia-(1) Induction: 1% propofol 1.5–3 mg/kg with 2% lidocaine 40 mg pretreatment, rocuronium 6 mg/kg; (2) Maintenance: inhaled anesthetics (desflurane) with Bispectral index (BIS) monitoring, remifentanil continuous infusion 0.1–1 ug/kg/min; (3) Recovery: Fentanyl 50–100 ug iv after turned off the remifentanil infusion. (with neuromuscular recovery using pyridostigmine and glycopyrrolate) ** 0.5% Bupivacaine 5 mL + normal saline 4 mL + epinephrine (0.1 mg/mL) 1 mL *** PRN—Tramadol 50 mg IV, Pethidine 50 mg IV **** Routine fluid—Acupan 60 mg IV, Marobiven 6.6 ml IV, Methocarbamol 3 g IV, Myraxan 30 mg IV, Traumeel 6.6 mg IV, Nasea 0.6 mg IV.

**Table 2 jcm-09-01087-t002:** Baseline data and clinical characteristics.

		MMA (n = 34)	Before Matching		After Matching	
			PCA (n = 100)	*P*-Value	PCA (n = 66)	*P*-Value
Age (years) ^a^		62.8 ± 11.0	67.1 ± 11.8	0.043*	64.5 ± 11.7	0.338
Sex ^b^	Male	17 (50.0%)	41 (41.0%)	0.360	29 (43.9%)	0.565
	Female	17 (50.0%)	59 (59.0%)		37 (56.1%)	
Smoking ^b^		6 (17.6%)	18 (18.0%)	0.963	9 (13.6%)	0.595
BMI (kg/m^2^) ^a^		24.6 ± 2.7	27.0 ± 3.9	0.001*	25.4 ± 3.4	0.346
Pain position ^b^				0.803		0.784
Axial		4 (11.8%)	9 (9.0%)		5 (7.6%)	
Radiating to leg		15 (44.1%)	50 (50%)		31 (47.0%)	
Combined		15 (44.1%)	41 (41%)		30 (45.5%)	
Preoperative NRS ^a^		7.5 ± 1.1	7.4 ± 1.0	0.375	7.2 ± 1.0	0.075
Preoperative ODI ^a^		23.6 ± 4.9	24.1 ± 5.2	0.659	24.8 ± 4.2	0.762
Preoperative medication ^c^				1.000		0.999
None		22 (64.7%)	62 (62.0%)		44 (66.7%)	
AAP/NSAIDs		2 (5.9%)	7 (7.0%)		3 (4.5%)	
Weak opioids		3 (8.8%)	9 (9.0%)		6 (9.1%)	
Strong opioids		1 (2.9%)	4 (4%)		3 (4.5%)	
Combination		6 (17.6%)	18 (18.0%)		10 (15.2%)	
ASA class ^a^		1.8 ± 0.5	1.8 ± 0.6	0.966	1.8 ± 0.6	0.858
AIBG (No.) ^b^		5 (14.7%)	22 (22.0%)	0.360	14 (21.2%)	0.432
Suction drain use (No.) ^b^		6 (17.6%)	22 (22.0%)	0.590	11 (16.7%)	0.902
Intraoperative complication ^c^		3 (8.8%)	6 (6.0%)	0.692	3 (4.5%)	0.406
Operation time (min) ^a^		220.8 ± 53.0	222.5 ± 51.7	0.878	220.4 ± 48.1	0.985

MMA = multimodal analgesia; PCA = patient-controlled analgesia; BMI = body mass index; NRS = numeric rating scale (0–10); ODI = Oswestry Disability Index; ASA = American Society of Anesthesiologists; AIBG = autologous iliac bone graft; ^a^ Mann–Whitney U test; ^b^ Pearson chi-squared test; ^c^ Fisher’s exact test; * indicates statistical significance.

**Table 3 jcm-09-01087-t003:** Postoperative outcomes.

		MMA (n = 34)	PCA (n = 66)	*P*-Value ^a^
Mean NRS	POD 0	5.7 ± 1.9	5.4 ± 1.8	0.302
	POD 1	4.2 ± 1.9	3.8 ± 1.5	0.282
	POD 2	3.4 ± 1.8	3.2 ± 1.5	0.901
Additional opioids consumption (ME/day)	POD 0	2.7 ± 6.7	5.1 ± 8.3	0.137
	POD 1	2.2 ± 5.9	8.2 ± 16.1	0.059
	POD 2	1.5 ± 4.8	7.7 ± 14.5	0.014*
Additional non-opioids analgesia consumption (mg/day)	POD 0	54.4 ± 81.1	37.1 ± 59.7	0.423
	POD 1	25.6 ± 49.6	51.5 ± 79.9	0.159
	POD 2	30.9 ± 65.2	34.8 ± 66.2	0.393

ME = morphine equivalents; POD = postoperative day; ^a^ Mann–Whitney U test; * indicates statistical significance *P* < 0.05.

**Table 4 jcm-09-01087-t004:** Side effects.

		MMA (n = 34)	PCA (n = 66)	*P*-Value
Nausea/Vomiting and Antiemetic drug use (%) ^a^	POD 0	16 (47.1%)	42 (63.6%)	0.112
	POD 1	16 (47.1%)	42 (63.6%)	0.112
	POD 2	9 (26.5%)	29 (43.9%)	0.088
Abdominal discomfort (%) ^b^	POD 0	9 (26.5%)	15 (22.7%)	0.577
	POD 1	8 (23.5%)	12 (18.2%)	0.942
	POD 2	1 (2.9%)	6 (9.1%)	0.417

POD = postoperative day; ^a^ Pearson chi-squared test; ^b^ Fisher’s exact test.

**Table 5 jcm-09-01087-t005:** Length of hospital stay, cost for pain control, and subjective satisfaction outcome.

	MMA (n = 34)	PCA (n = 66)	*P*-Value
Length of hospital stay ^a^ (day)	7 (5–8)	8 (7–11)	0.001*
Total Cost (USD) ^b^	70.6 ± 0.9	173.4 ± 3.3	<0.001*
Routine use			
Pump setup		80.9 ± 0.0	
Pump PCA per day		15.1 ± 3.2	
Pump medication		76.5 ± 0.0	
MMA medication	66.9 ± 0.0		
Rescue medication			
Tramadol ^b^	0.6 ± 0.8	0.7 ± 0.8	0.315
Opioid (Demerol) ^b^	0.1 ± 0.2	0.2 ± 0.3	0.022
Subjective satisfaction (1–5) ^b, c^	2.5 ± 0.8	2.2 ± 0.7	0.108

^a^ Mann–Whitney U test and summary statistics are presented as median (IQR); ^b^ Student t-test; ^c^ 1: Very satisfied, 2: Somewhat satisfied, 3: Neutral, 4: Somewhat dissatisfied, 5: Very dissatisfied; * indicates statistical significance *P* < 0.05.

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
