# Peer review of "Multimodal Analgesia (MMA) Versus Patient-Controlled Analgesia (PCA) for One or Two-Level Posterior Lumbar Fusion Surgery"

_jcm, 2020, doi:10.3390/jcm9041087_

Round 1

Reviewer 1 Report

I found this paper very nice, and with some very interesting data to reduce costs in hospitals and the use of opioids. With the changes made, the work has improved substantially. So I do not suggest any further modifications.

Author Response

We appreciate your thoughtful suggestions and insights, which have enriched this manuscript and produced a more balanced and better account of the research.

Reviewer 2 Report

congrats! It´s a well designed article about multimodal perioperative analgesia. but you may explain why don´t you consider other several factors in perioperative pain (such as: chronic treatments, depresion state, ...)

Author Response

Thank you for your comment.
We appreciate your thoughtful suggestions and insights, which have enriched this manuscript and produced a more balanced and better account of the research.

1) This study was conducted in patients with degenerative spinal disorder who underwent 1-2 segments fusion. In Korea, spinal fusion surgery for degenerative spinal disorder can be performed in patients with a sufficient history of more than 3 months of conservative pain treatment. Therefore, every patient was treated for more than three months of conservative treatment.

We added this in the text. Line 76-77

--> All patients who had a history of more than three months of conservative pain control underwent 1 or 2 levels posterior lumbar fusion (PLF) with degenerative spinal disorder.

2) This study was conducted in patients with degenerative spinal disorder who underwent 1-2 segments fusion. Preoperative evaluation such as medical history, current medication, and previous pain control method of spine disease by our medical team was performed, allow no exceptions, in every patient.

If they had a history of depressive disorders or took antidepressant medication, we consulted the patient to our psychological department, and we did not take a spine surgery as much as possible and we led the patient to conservative treatment.

In conclusion, patients with depressive disorder were excluded in this study.
We added this in the exclusion. Line 69.

--> The exclusion criteria were: any history of prior lumbar spine surgery, hepatic dysfunction, renal insufficiency, depressive disorder, an emergency surgery, a history of drug/alcohol abuse, or prior adverse or allergic reactions to any of the analgesic medications to be administered (celecoxib, pregabalin, acetaminophen or oxycodone).

Reviewer 3 Report

Thanks for addressing my comments. The manuscript has improved significantly.

Author Response

(The authors gave the same response as above.)
